A novel deep learning approach for predicting stone-free rates post-ESWL on uncontrasted CT

Efiloglu Ozgur 1
http://orcid.org/0000-0003-1866-4721 Yildirim Muhammed 2
Yildirim Kadir 3
http://orcid.org/0000-0001-5071-4616 Bingol Harun 4 harun.bingol@ozal.edu.tr
Akalin Mustafa Kaan 1
Culpan Meftun 1
http://orcid.org/0000-0002-3513-0329 Alatas Bilal 5
Yildirim Asif 1
1 Urology, Istanbul Medeniyet University , Istanbul , Turkey
2 Computer Engineering, Malatya Turgut Ozal University , Malatya , Turkey
3 Urology, Elazig Fethi Sekin City HTRC , Elazig , Turkey
4 Software Engineering, Malatya Turgut Ozal University , Malatya , Turkey
5 Software Engineering, Firat (Euphrates) University , Elazig , Turkey
Wan Shibiao
Electronic publication date: 2025 Aug 11
Publication date: 2025
Volume: 11
Electronic Location ID: e3111
Received 2024 Dec 12; Accepted 2025 Jul 18
Copyright: © 2025 Efiloglu et al.
Copyright year: 2025
Copyright holder: Efiloglu et al.
License: This is an open access article distributed under the terms of the Creative Commons Attribution License, which permits unrestricted use, distribution, reproduction and adaptation in any medium and for any purpose provided that it is properly attributed. For attribution, the original author(s), title, publication source (PeerJ Computer Science) and either DOI or URL of the article must be cited.
License URL: https://creativecommons.org/licenses/by/4.0/

Keywords: Artificial Intelligence, ESWL, HOG, Kidney Stone, LBP

Funding: The authors received no funding for this work.

==============================
Extracorporeal shock wave lithotripsy (ESWL) is one of the most often employed therapy methods for managing kidney stones. In our work, we sought to assess the efficacy of the artificial intelligence model developed using non-contrast computed tomography (CT) images in predicting stone-free rates for ESWL. The main difference between this study and other studies is that it proposes an artificial intelligence-based model that predicts the success of ESWL treatment using artificial intelligence methods. Data from 910 patients who underwent ESWL between January 2016 and June 2021 were analyzed retrospectively. Since the local binary pattern (LBP) and histogram of oriented gradients (HOG) feature extraction methods gave more successful results than other methods, a new feature map was obtained using the neighborhood component analysis (NCA) dimension reduction method after combining the features obtained using these methods. Then, the reduced feature map was classified into classifiers. In conclusion, we analyzed the effect of ESWL treatment using different artificial intelligence methods and found that the prediction accuracy was 94% on average. Results were obtained from seven different convolutional neural networks (CNNs) and two textural-based models in the study. Since textural-based models achieved the highest success among these models, these models were used as the base in the proposed model. The proposed model achieved better results than nine different models used in the study. When the results obtained from the proposed hybrid model for ESWL prediction are examined, this model will guide experts in the treatment of the disease.

Introduction

ESWL, which is a non-invasive intervention in the treatment of urinary system stones, still has an important role in current urology practice. Chaussy et al. (1984) published their ESWL experience for the first time in 1984. With the technological development of ESWL devices over the years, complication rates have decreased and success rates have increased. In the conditions of the last pandemic, the complications of surgery and anesthesia have increased due to COVID-19. Postponing non-emergency procedures and encouraging patients to treatments such as ESWL, which do not require general anesthesia, have come to the fore (Collaborative, 2020; Ribal et al., 2020).

The European Association of Urology guidelines recommends that ESWL be considered among the first-line treatment options in the treatment of kidney stones smaller than 2 cm. The success of ESWL may be influenced by factors such as stone size, position, patient behaviors, narrow infundibulopelvic angle, long calyx, long stone skin distance (TCM), thin infundibulum, and shock wave resistant stones (calcium oxalate monohydrate, brushite, or cystine) (Akram et al., 2024). In our study, we aimed to determine the success of the non-contrast CT artificial intelligence model in predicting stone-freeness for ESWL success.

In recent years, artificial intelligence (AI) techniques have been increasingly integrated into urological practice. These studies offer significant potential to improve patient outcomes in ESWL procedures. AI studies have been used to predict treatment success based on patient-specific and stone-related parameters such as stone size, location, density, and skin-stone distance. In some of these studies, automatic diagnosis of ureteral stones and hydronephrosis grading have been performed (Chakit et al., 2023; Bugday et al., 2023; Yang et al., 2024). In general, studies that predict ESWL are quite rare in the literature. Therefore, this study conducted for ESWL prediction is of great importance. Studies based on AI for ESWL prediction will help clinicians choose the most appropriate procedures for ESWL. In this way, the procedure will increase efficiency and unnecessary interventions will be reduced. As personalized medicine gains importance, the use of AI in optimizing the ESWL decision-making process will produce promising results in modern urology. In this study, we used non-contrast computerized tomography (CT) images to predict stone-free status for ESWL success.

In the study, results were obtained using seven different CNNs and two different textural-based architectures on the CT images in the dataset. A hybrid model was developed for ESWL estimation. The most successful results were produced by textural-based local binary pattern (LBP) and histogram of oriented gradients (HOG) architectures. In the first step of the proposed model, the feature maps obtained with these architectures were combined. In this way, different features of the same image were used together. In the second step of the proposed model, the neighborhood component analysis (NCA) method was used to eliminate unnecessary features and to make the model produce more successful results. Thanks to NCA, the size of the feature map was reduced, and the training time of the proposed model was shortened. Finally, the feature map optimized with the NCA method was classified in six different machine learning classifiers. As a result of the comprehensive comparisons, support vector machine (SVM), known for its superior generalization ability, especially in complex and multi-dimensional data sets, provided the highest accuracy rate compared to other classifiers and played a decisive role in the success of the model. The accuracy value of the proposed model was 94%. This ratio shows that the proposed model has high reliability in predicting ESWL treatment success and can be used as an effective tool in clinical decision support systems.

Material and method

In this part of the study, information is given about the data set obtained from non-contrast abdominal CT images, HOG and LBP methods, which are textural feature extraction methods, and NCA, which is the feature size reduction method. In addition, the classifiers used in the model we have proposed in this study are examined. Figure 1 depicts the fundamental diagram of our suggested concept.

Figure 1 Basic diagram of the developed model.

Dataset

Data from 910 patients who underwent ESWL between January 2016 and June 2021 were analyzed retrospectively. A Lithostar Modularis Lithotripter (Siemens Medical Systems, Erlangen, Germany) was used for treatment. All patients were diagnosed radiographically with non-contrast CT before treatment. Ministry of Health Istanbul Civilization University Goztepe Training and Research Hospital Clinical Research Ethics Committee gave the study its permission (approval number: 2021/0610). This retrospective study was exempt from the requirement to obtain informed consent from participants. ESWL was not performed in patients with active urinary tract infection, anatomical anomaly, uncontrolled bleeding diathesis, or distal obstruction. Multiple kidney stones and patients under 18 years of age were excluded from the study. Pre-ESWL urinalysis, complete blood count, and bleeding and coagulation times of all patients were evaluated.

ESWL energy level protocol in the study: 0.1 Joule (J) between 0–100 shock waves, it was increased gradually between 100–800 shock waves, and 1.0 J energy was made in 800 shock waves. From that point forward, the energy was increased by 0.3 J for every 150 shockwaves. A maximum of 3,000 shock waves or 3.5 J energy was used until stone fragmentation was seen on fluoroscopy. ESWL was performed without anesthesia. Intramuscular diclofenac sodium 75 mg was administered as an analgesic treatment before ESWL.

Direct urinary system radiography, ultrasonography, or non-contrast CT (for radiolucent stones) were checked 21 to 30 days after the ESWL session. Stone disappearance or leftover bits of less than 4 mm were considered successful. Patients who did not have stone clearance and/or did not decrease in stone size after the first three sessions were defined as failures and were referred to other treatment alternatives. Examples of CT images of the abdomen used in the study are given in Fig. 2.

Figure 2 Abdomen tomography image samples.

The ESWL+ group describes the abdominal tomography images of patients who underwent ESWL and benefited from this treatment. The ESWL- group defines the abdominal tomography images of patients who underwent ESWL and did not benefit from this treatment.

Textural feature extraction methods, neighborhood component analysis and classifiers

Deep learning approaches have become increasingly popular in the field of biomedicine in recent years, owing to their high accuracy in disease image classification. However, deep learning methods have negative aspects as well as beneficial aspects. At the beginning of these disadvantages, the models spend a lot of time classifying the images. During our tests, we suggested a new mathematical model to solve this problem by using textural feature extraction approaches. The most widely utilized LBP and HOG methods were used in this research (Demir et al., 2022; Ojala, Pietikäinen & Mäenpää, 2001; Dalal & Triggs, 2005).

LBP is a basic yet effective texture operator that identifies an image’s pixels by delimiting each pixel’s neighborhood and treating the output as a binary number. The LBP tissue operator has been a prominent technique in the biomedical industry because to its discrimination power and computational convenience. LBP has a simple structure that can extract low-level features by separating them and is shown in Eq. (1).

(1) LBPX,Y=∑x=0X−1⁡s(gx−gc)2x,s(n)={1,n≥00,n<0.

In Eq. (1), the gray value of the center pixel is gc. gx shows the values of the neighbors of the central pixel. X denotes the total number of related neighbors and Y denotes the radius of the neighborhood. Assuming that the coordinate of gc is (0, 0), the coordinates of gx are (Ycos(2πxX),Ysin(2πxX)). Interpolation is used to estimate the gray values of neighbors who are not in the image grids (Guo, Zhang & Zhang, 2010).

Over other identifiers, the HOG offers a few advantages. It does not change in geometric or photometric modifications because it works on local cells, save for object orientation. Larger spatial regions are the only places where such changes occur.

The best of the features obtained using LBP and HOG methods are selected using the distance metric method. The distance metric has a very important place in pattern recognition applications. Many metric learning methods are available in the literature. One of the most successful metric learning algorithms is NCA. With NCA, the size reduction process is performed by keeping the useful features and removing the unhelpful features (Yang, Wang & Zuo, 2012).

Classification can be performed by giving the most useful features obtained by NCA method to the original data set consisting of disease images obtained from real patients, to different classifiers. There are many classification algorithms in the literature. In the study, SVM, k-nearest neighbors (kNN), discriminant analysis (DA), decision tree (DT), ensemble suspace (ES) and naive Bayes (NB) classifiers were used during the experiments (Suykens & Vandewalle, 1999; Dudani, 1976; Klecka, Iversen & Klecka, 1980; Laurent & Rivest, 1976; Kuncheva et al., 2010; Rish, 2001).

Proposed model

Using the dataset individually, the LBP and HOG methods were used to produce feature maps for the generated model. The LBP approach produced a feature map that is 910 × 2,891 in size. The size of the feature map obtained by the HOG method is 910 × 1,296. These feature maps obtained with LBP and HOG were combined. The size of the combined feature map is 910 × 4,187. Applying the NCA approach to this acquired feature map reduces the size of the feature map. The size of the reduced feature map is 910 × 500. The size of the feature map obtained in the model we proposed has been reduced by 88.05%. Thus, the computational complexity problem, which is very common in image processing applications, has been largely overcome. In this way, not only a model with high accuracy has been proposed, but also the computational complexity problem has been reduced to a minimum. Finally, the reduced feature map was given to different classifiers and the classification process was carried out. Figure 3 provides the created model’s diagram.

Figure 3 The way the developed model works.

In addition, experiments were conducted with seven different CNN architectures in the study and the results of these architectures were compared with the proposed model. The findings showed that the hybrid model created with classical texture-based methods such as LBP and HOG provided higher classification accuracy compared to CNN-based models. This was observed to provide more effective solutions as an alternative to CNN architectures, especially in cases where texture information is at the forefront, such as CT images, in ESWL estimation.

Statistical analysis

In the study, models based on artificial intelligence were used to categorize non-contrast CT images. The study measured and compared the effectiveness of the suggested techniques using a variety of indicators. The confusion matrix is used to compute the evaluation metrics for the study. Figure 4 illustrates a confusion matrix as an example.

Figure 4 Confusion matris.

In the confusion matrix given in Fig. 4, it is seen that there are two classes as ESWL+ group and ESWL- group.

True positive (TP): The total number that the model predicts as ESWL+, which is actually ESWL+,

False positive (FP): The total number that the model predicts as ESWL− is actually ESWL+,

False negative (FN): The total number that the model predicts as ESWL+, which is actually ESWL−,

True negative (TN): The total number that the model predicts is ESWL−, which is actually ESWL−.

The study evaluated the performance of models based on artificial intelligence using the evaluation measures listed in Table 1.

Table 1 Performance metrics.

Metrics	Equations	
Accuracy (ACC)	(TP + TN)/(TP + TN + FP + FN)	
Specificity (SPE)	TN/(FP + TN)	
Sensitivity (SENS)	TP/(TP + FN)	
Precision (PRE)	TP/(TP + FP)	
Negative Predicted Value (NPV)	TN/(FN + TN)	
False Discovery Rate (FDR)	FP/(FP + TP)	
False Positive Rate (FPR)	FP/(FP+ TN)	
False Negative Rate (FNR)	FN/(FN + TP)	
F1-score (F1)	2TP/(2TP + FN + FP)	
Matthews Correlation Coefficient (MCC)	TP * TN-FP * FN/sqrt((TP + FP) * (TP + FN) * (TN + FP) * (TN + FN))	

Performance evaluation metrics used in classification problems are important in terms of evaluating the success of models comprehensively and objectively. Since the accuracy value alone may not fully reflect the sensitivity of the model to different types of errors, it is important to evaluate with more than one metric. It can analyze the weaknesses and strengths of the models in detail. These metrics allow us to detect misleading results, especially in unbalanced data sets. In addition, by providing consistent and comparable criteria in model comparisons, it increases the scientific reliability and validity of the results obtained.

Results

In the study, the performance of many artificial intelligence models was tested. The most appropriate model for the ESWL treatment method has been discussed. As a result, it has been observed that LBP and HOG feature extraction methods give more successful results compared to other methods. In the study, firstly, the test results obtained by using LBP and HOG methods are given. Then, after combining the features obtained using the LBP and HOG methods, a new feature map was obtained using the NCA dimension reduction method. Thus, a feature map consisting of the most useful features was obtained. Finally, the resulting reduced feature map was classified into classifiers. In addition to these methods, the performances of pre-trained models were also examined in the study and compared with the proposed model. In the study, the cross-validation process was applied to prevent overfitting. In addition, the same metrics were used to make comparisons under equal conditions in all classifiers and models.

The accuracy values of the feature maps obtained by using LBP and HOG methods in six different classifiers are given in Table 2.

Table 2 Accuracy values of LBP and HOG models in classifiers.

Methods	Accuracy rate	
DT	DA	NB	SVM	kNN	ES	
LBP	72.5	69.5	58.9	90.4	90.7	91.4	
HOG	74.3	73.1	62.9	91.5	92.1	91.9	

As can be seen in Table 2, the classifier in which the LBP method is the most successful is the ES classifier, while the classifier in which the HOG method is the most successful is the kNN classifier. Confusion matrices obtained in the LBP classifier are given in Fig. 5.

Figure 5 Confusion matrices obtained in different classifiers using the LBP method.

When Fig. 5 and Table 2 are examined, it is seen that the classifier in which the LBP feature extraction method is most successful is the ES classifier. The ES classifier classified 832 of 910 abdominal CT images correctly and 78 of them incorrectly. ES correctly predicted 424 abdominal CT images from 456 ESWL+ patients and incorrectly predicted 32 of them. Similarly, ES predicted 408 abdominal tomography images obtained from 454 ESWL- patients correctly and 46 of them incorrectly. The LBP method achieved the highest accuracy with 91.4% in the ES classifier.

Another feature extraction method used in the study is the HOG feature extraction method. Confusion matrices obtained in different classifiers using the HOG feature extraction method are shown in Fig. 6.

Figure 6 Confusion matrices obtained in different classifiers using the HOG method.

Figure 6 and Table 2 analysis reveal that the kNN classifier is the one in which the HOG feature extraction method performs best. kNN classifier classified 836 of 910 abdominal CT images correctly and 74 of them incorrectly. kNN correctly predicted 424 abdominal tomography images from 456 ESWL+ patients and incorrectly predicted 32 of them. Similarly, kNN correctly predicted 412 abdominal tomography images obtained from 454 ESWL− patients and incorrectly predicted 42 of them. The HOG method has the highest accuracy value of 92.1% in the kNN classifier.

Finally, Table 3 shows the accuracy values obtained in the suggested model for ESWL treatment. The proposed model was shown to produce better outcomes than the other models utilized in the investigation.

Table 3 The accuracy values obtained in the proposed model.

Methods	Accuracy rate(%)	
DT	DA	NB	SVM	KNN	SE	
Proposed model	73.8	75.8	58.9	94	89.9	89.7	

When looking at Table 3, it is seen that the most successful classifier is the SVM classifier with 94%. SVM classifier is followed by kNN with 89.9%, ES with 89.7%, DA with 75.8%, DT with 73.8% and NB with 58.9%, respectively. It would be more accurate to use the SVM classifier in the proposed model for ESWL treatment.

Figure 7 shows the confusion matrices that were obtained using the proposed approach.

Figure 7 Confusion matrices obtained in the proposed model.

When Fig. 7 is inspected, it becomes clear that the proposed model’s feature map is divided into six separate classifiers. NB is the classifier that performs the least well out of all of them. 536 of the 910 abdominal CT images were accurately identified by the NB classifier, while 374 were wrongly classified. The most successful classifier used in the study is SVM. SVM estimated 855 correctly and 55 incorrectly from 910 abdominal tomography images. SVM correctly predicted 422 abdominal CT images from 456 ESWL+ patients and incorrectly predicted 34 of them. Similarly, SVM correctly predicted 433 abdominal tomography images from 454 ESWL− patients and incorrectly predicted 21 of them. Our proposed model achieved the highest accuracy of 94% in the SVM. The area under curve (AUC) curves acquired in this classifier are given in Fig. 8.

Figure 8 AUC curves obtained in the proposed model.

When Fig. 8 is examined, the receiver operating characteristic (ROC) curve shows the relationship between the true positive rate and false positive rate at different threshold values of the model. AUC value was calculated as 0.98 for the ESWL+ and ESWL– classes. These results show that the proposed model has high discrimination power and can make a strong distinction between the classes. In the SVM, it has been found that the constructed model achieves the maximum accuracy rate of 94%. Since it would not be correct to evaluate the performance of a model on only one metric, other performance criteria are also given in Table 4.

Table 4 Performance indicators of the proposed model (%).

MCC	NPV	FPR	SENS	PRE	
87.95	95.37	7.28	95.26	92.54	
F1	FDR	FNR	ACC	SPE	
93.88	7.46	4.74	93.96	92.72	

When looking at Table 4, it is clear that the proposed model is effective. Therefore, the proposed model can predict ESWL treatment success. Thus, the workload of the urologist can be reduced, preventing unnecessary treatment to the patient and allowing the patient to receive more accurate treatment.

Discussion

ESWL has an important place among the treatment options due to its ease of use, non-invasiveness, and high effectiveness in upper urinary tract stones. In addition to clinical parameters such as age, gender, and BMI that affect the success of ESWL, CT parameters such as stone location, number, diameter, Hounsfield density of the stone, TCM, and presence of hydronephrosis are available (Kim et al., 2016). Clinical nomograms are used to help urologists better identify ideal patients for ESWL (Tran et al., 2015). Numerous studies have been reported in the literature to determine the factors affecting the stone-free rate using statistical methods (Abdelhamid et al., 2016; Waqas et al., 2018; Shinde et al., 2018).

In the treatment of kidney stones, ineffective procedures can be avoided by evaluating the response to ESWL before the procedure and choosing better treatment methods for treatment management. Understanding the variables that determine ESWL success will increase its efficiency and assist avoid needless treatments.

The practical implications of the proposed model are particularly important in reducing unnecessary ESWL procedures. The proposed model accurately predicts the probability of treatment success, allowing clinicians to make more informed decisions when selecting appropriate procedures for ESWL. The proposed model not only improves treatment outcomes but also minimizes the risk of subjecting patients to ineffective procedures. This reduces patient burden, procedure costs, and potential complications. Furthermore, the proposed model can be integrated into clinical workflows to improve personalized treatment planning and contribute to more efficient use of healthcare resources.

It is recommended that AI applications be validated in an external dataset before they are publicly available. For this reason, a cross-country or nationwide database should be created and shared. Machine learning and artificial intelligence will play an important role in clinical decision-making in lithotripsy in the near future. Artificial intelligence is frequently used in the diagnosis of different diseases in the biomedical field. Classification of images of brain tumor (Ilani, Shi & Banad, 2025), Cholesteatoma (Ouattassi et al., 2025), kidney stones (Yildirim et al., 2021) and vesicoureteral reflux (Chen et al., 2025) are some of the main applications of artificial intelligence techniques.

The main difference of this study from other studies in the literature was that the disease classification process was not carried out using artificial intelligence methods. Here, an artificial intelligence-based model is proposed that can predict the success of ESWL treatment.

In this study, abdominal CT images of patients who underwent ESWL treatment were used. In the study, images of patients who responded positively to the treatment and did not respond positively to the treatment were scanned backward and labeled by five experts in their fields. The aim of this study is to determine whether ESWL treatment is necessary for the patient, rather than to classify any disease, and an artificial intelligence-based model is proposed. The proposed model has been compared with many artificial intelligence methods in the literature. In the experiments of pre-trained CNN models, hyper parameters were determined as solver name SGDM, BatchSize 10, Epochs 15, InitialLearnRate 1.0000e−04, Shuffle every-epoch. In addition, 80% of the CT images in the dataset were used for training the models and the rest for testing. The accuracy values of the artificial intelligence-based models used in the article are shown in Table 5.

Table 5 Comparison of the proposed model with other models (%).

Alexnet	Efficientnetb0	Googlenet	InceptionV3	MobilenetV2	
58.79	55.49	55.49	53.29	52.19	
Resnet50	Shufflenet	LBP+Classifiers	HOG+Classifiers	Proposed model	
57.14	52.74	91.4	92.1	94	
Note:

The proposed model is shown in bold.

When looking at Table 5, it can be shown that the suggested model outperforms other models that have been published. Alexnet, Efficientnetb0, Googlenet, InceptionV3, MobilenetV2, Resnet50, and Shufflenet models used in the study are pre-trained models. These models are frequently used in image classification studies. LBP and HOG are textural-based feature extraction methods. The proposed model is a hybrid model based on artificial intelligence. The difference from other models is that since feature maps are optimized by the NCA size reduction method, it will produce faster results.

This study has some limitations. Although the data are kept prospectively, the study was planned retrospectively. Second, the study was conducted with a single center and a single ESWL device. Another limitation is the lack of metabolic evaluation data for the patients. The strength of the study is that when the literature is examined, no study has been found on the classification of ESWL treatment methods with artificial intelligence models. In addition, it is among our aims to obtain data from different centers, develop a more complex system and make this system available online.

Conclusions

In this study, the effectiveness of ESWL treatment was analyzed with the proposed hybrid model. In the hybrid model, where the feature maps of radiological images were combined with the NCA dimensionality reduction technique and machine learning classifiers, an accuracy value of 94% was achieved. This accuracy rate reveals the potential of the model to guide clinicians in determining patients who may benefit from ESWL treatment, thus contributing to the reduction of unsuccessful treatment applications and unnecessary interventions. However, the sample size of the study and the limitation of single-center data limit the generalizability of the findings. Therefore, multicenter studies covering larger patient groups should be conducted in order to evaluate the validity and clinical applicability of the proposed model.

Supplemental Information

Supplemental Information 1 Code.

Supplemental Information 2 Raw data.

Additional Information and Declarations

Competing Interests

Bilal Alatas is an Academic Editor for PeerJ.

Author Contributions

Ozgur Efiloglu conceived and designed the experiments, performed the experiments, analyzed the data, authored or reviewed drafts of the article, and approved the final draft.

Muhammed Yildirim conceived and designed the experiments, performed the experiments, analyzed the data, performed the computation work, prepared figures and/or tables, and approved the final draft.

Kadir Yildirim conceived and designed the experiments, analyzed the data, authored or reviewed drafts of the article, and approved the final draft.

Harun Bingol conceived and designed the experiments, performed the experiments, performed the computation work, prepared figures and/or tables, and approved the final draft.

Mustafa Kaan Akalin conceived and designed the experiments, performed the experiments, prepared figures and/or tables, authored or reviewed drafts of the article, and approved the final draft.

Meftun Culpan conceived and designed the experiments, performed the experiments, analyzed the data, authored or reviewed drafts of the article, and approved the final draft.

Bilal Alatas conceived and designed the experiments, analyzed the data, performed the computation work, prepared figures and/or tables, and approved the final draft.

Asif Yildirim conceived and designed the experiments, performed the experiments, authored or reviewed drafts of the article, and approved the final draft.

Ethics

The following information was supplied relating to ethical approvals (i.e., approving body and any reference numbers):

The Ministry of Health Istanbul Civilization University Goztepe Training and Research Hospital Clinical Research Ethics Committee approved the study (2021/0610).

Data Availability

The following information was supplied regarding data availability:

Code and raw data are available in the Supplemental Files.

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
