# Peer review of "A novel deep learning approach for predicting stone-free rates post-ESWL on uncontrasted CT"

_PeerJ Computer Science, doi:10.7717/peerj-cs.3111_

## Round 0.1 · original submission · Major Revisions

·

Basic reporting

1. Background and references are adequate, but missing references to the latest advancements in AI-based prediction models.
2. Figures and data are relevant and well-labeled, but captions need more detail.

Experimental design

1. Research question is clearly defined and relevant, addressing an important gap in ESWL prediction.
2. Methodology is detailed but could benefit from a pseudocode or flowchart for reproducibility.

Validity of the findings

1. Great statistical analysis, but lacking confidence intervals and broader comparisons.
2. Needs validation on external, multi-center datasets.

Additional comments

This is a great article, this has potential for real world application, and nicely written. Please see below for comments.

Abstract:
The abstract is concise and provides an overview of the study

1. Add a specific statement highlighting how the proposed combination of LBP, HOG, and NCA for ESWL prediction outperforms previous methods.
2. Rephrase to focus on novelty of the approach - “The artificial intelligence method will be helpful to guide ESWL treatment selection”


Introduction:
The introduction sets the context well.

1. Provide a detailed comparison of the limitations of existing models for ESWL success prediction and explain why LBP, HOG, and NCA were selected.
2. Add references to recent studies (e.g., 2022–2024) on AI-based medical imaging models for a stronger literature context.
3. Clearly state the specific research gap that this study addresses and the novelty of integrating feature extraction and dimensionality reduction methods with classifiers.


Methods:
The methods section provides sufficient technical details

1. Consider including a pseudocode or detailed workflow for the proposed model in the Proposed Model subsection to aid reproducibility.
2. Provide additional details on the rationale for combining LBP and HOG as feature extractors and how their outputs were optimized by NCA.
3. Can you clearly explain why specific classifiers (SVM, kNN, etc.) were chosen and their hyperparameter tuning process.




Results
The results are well-presented

1. Can you Include confidence intervals for the accuracy metrics presented in Tables 2 and 3 to strengthen statistical reliability.
2. Can you provide a more detailed explanation of the confusion matrices in Figures 5–7, especially highlighting why specific classifiers performed better or worse.
3. It would be better to compare the proposed model's results with more pre-trained models beyond AlexNet, such as recent Transformer-based architectures.


Discussion:
The discussion is insightful but could expand.

1. Can you expand on the practical implications of the model, particularly its impact on reducing unnecessary ESWL procedures.
2. Can you discuss potential biases introduced by the dataset, such as the use of a single ESWL device and single-center data.


Limitations:
The limitations section is briefly mentioned and lacks detail

1. You may have to clearly state the limitations of the study, such as the retrospective design and the lack of external validation.
2. Can you discuss the potential overfitting risks due to the small dataset size and suggest mitigation strategies.


Practical Applications:
Practical applications are implied but not explicitly discussed

1. Can you please include a discussion on how the model could be integrated into clinical workflows, such as in radiology or urology departments.
2. Can you suggest potential real-time applications, such as automated pre-ESWL treatment recommendations based on CT scans.

Reviewer 2 ·

Basic reporting

The study is very complex and should also be reviewed by AI or programming experts. The article is well written with a good flow. The introduction part should be elaborated on AI and other machine learning role in medical, especially urology. The novelty of this study is better shown in the introduction. The tables are too many, some of them may be combined. Some abbreviation should be first explained in the introduction as it was first mentioned in this part. I suggest line 47-51 should not be mentioned as it is uncommon to make an outline in the end of the introduction part.

Experimental design

The aim is clear for this study but the parameters are not well mentioned. This study should show sensitivity, specificity, PPV, and NPV as it predicts an outcome. The author should also mentioned what parameters is used to identify the outcomes, like HU. This study should show the subject characteristics that necessary to show the basic subjects data. This may be useful for the future study to analyze if any results are different.

Validity of the findings

Some results are redundant, the result should be only focusing on the highlighted data. The AUC should also be added with R square and hosmer lemeshow to show the applicability of the AUC. The author should also discuss about the superiority of this prediction model compared to other model based on the sensitivity, specificity, NPV, PPV, accuracy, precision as it is the basic parameters. The opening in first sentence of the conclusion is unnecessary, the conclusion should focus on the result so the reader could instantly catch reasons to use this prediction model.

Reviewer 3 ·

Basic reporting

to this reviewer, the manuscript does not meet the basic standards of clarity, conciseness and technical accuracy required. at a minimum, professional editing by a scientific writer.

Several citations are inappropriate or flat out completely uncorrelated with the sentence where they are used. The most egregious one is probably reference #4

The tables are difficult to intepret, as they lack essential details that are not provided in the caption or text either.

Experimental design

This is the weakest area in my opinion. While the results seem on paper quite compelling, this section lack any meaningful detail to help a reader attempt to reproduce them. Specifically, no results are provided about the CT imaging protocols, whether the algorithm is applied to the whole volume or just the kidney, any processing of the data or how the models are trained. The fundamental premise of a scientific article is to provide a reader with some insight on how to reproduce the data, and this manuscript fails at that, in my opinion.

Validity of the findings

There is definitely some novelty in the reported findings. Unfortunately, because of the limitations listed in the methods, it is hard to assess how robust and scientifically sound these results are.

·

Basic reporting

The structure of the manuscript is very clear. It is very well written and very easy to follow and understand. However, I have several comments and a few ideas to strengthen the paper. I recommend the following revisions:
In Introduction section
- ESWL, which is a non-invasive intervention in the treatment of urinary system stones, still has an important role in current urology practice.
This should be more explained as a widely used method by the worldwide population. High rate of free stone… with appropriate references (https://doi.org/10.1007/s00240-023-01407-9),
The hypothesis is neutral. Authors should present a clear hypothesis about the study. the importance of information technology in the medical field and specifically Urolithiasis treatment by ESWL. A paragraph about its implementation in this purpose regarding previous studies is required.

Experimental design

In introduction section, Authors consider indicating the question research.
I suggest a study design subsection to present the experimental design, population origin and characteristics and methods adopted.
inclusion and exclusion criteria should be defined.
Statistical analysis is well described.

Validity of the findings

The results are clearly presented and described. However, they need more discussion.
The discussion provides a good interpretation of the results but lacks integration with relevant literature. More references to previous studies supporting or contrasting the findings would strengthen the discussion.
- Other limitations should be mentioned as the size of the population, statistical methods used…and future directions.
The conclusion could be more impactful if it included concrete clinical recommendations based on the study's findings. It would also be relevant to add suggestions for future research.

Additional comments

Please ensure that every reference follow the referencing style used by this journal.

Reviewer 5 ·

Basic reporting

The manuscript titled “A Novel Deep Learning Approach for Predicting Stone-Free Rates Post-ESWL on Uncontrasted CT” by Efiloglu et al. presents a hybrid machine learning model that predicts post-ESWL stone-free rates from uncontrasted CT images using LBP, HOG, and NCA feature extraction. I am recommending a rejection for this manuscript for several reasons stated below:

Strong points:
1. The manuscript addresses an important clinical problem of predicting stone-free outcomes following ESWL which could significantly aid clinical decision-making and reduce unnecessary procedures.
2. The study uses a relatively large dataset of 910 patients, lending credibility and potential robustness to the results.
3. The study claims to have good performance of the dataset.

Weak points (Basic reporting):

1. The quality of the figures are extremely poor.
2. No figure captions? The captions for the figures should be updated with more detailed explanations. This is basic in scientific writing.
3. More figures needed to explain results visually.
4. The table qualities are equally poor. The table designs need to be more efficient. This is basic stuff in scientific writing.
5. The overall writing quality is also poor. The authors need to dig into how scientific writing is generally done.
6. The introduction section is too small. It needs to discuss more to let the reader understand the problem deeply. Also need to discuss proper settings of the models.
7. The literature review part is basically non-existing. The authors need to discuss these in detail and show why their problem and method are interesting.
8. Limitation of the work should be discussed in detail.
9. The source code should be uploaded to github to verify reproducibility.
10. The abstract should be updated with more numerical results.

Experimental design

Weak points (Experimental design):

1. The combination of LBP, HOG, and classical classifiers is not innovative in the current state of the field. Deep learning models, if properly trained, typically outperform such handcrafted approaches. None of these methods are the authors’ own invention. No problem-specific architecture used like U-Net, or SAM.
2. The method section needs more details. Too in-general and superficial explanations.
3. The manuscript overstates the immediate clinical applicability of the model without acknowledging overfitting risks, limited diversity of data, and the absence of prospective or multi-center validation.

Validity of the findings

Weak points (Validity of the findings):
1. The manuscript briefly compares its method with deep learning baselines (e.g., AlexNet, ResNet50) but does not provide fair comparisons as physical insights are provided to back the results up.
2. The comparison with deep learning models (AlexNet, ResNet, etc.) is not robust. These models appear to have been used without sufficient tuning on the specific dataset, which unfairly disadvantages them against the hybrid feature extraction approach. Stronger comparisons with properly optimized and trained deep models are needed.
3. “...found that the prediction accuracy was 94% on average” - we know that solely accuracy is not a proper measure to compare for these tasks from basic deep learning.
4. Single-center, retrospective data without external validation undermines the reliability of the conclusions.

---

## Round 0.2 · accepted · Accept

Reviewers are satisfied with the revisions, and I recommend accepting this manuscript.

·

Basic reporting

This version looks better. Thank you for addressing the review comments.

Experimental design

-

Validity of the findings

-

·

Basic reporting

The changes made significantly improve the clarity and scientific rigor of the text. I have no further comments to add.

Experimental design

-

Validity of the findings

The modifications made significantly improve the clarity and scientific rigor of the text. All underlying data have been provided and discussed.

Reviewer 5 ·

Basic reporting

The authors have considered most of my suggestions.

Experimental design

-

Validity of the findings

-